# Preparation and Characterization of Nanoporous Activated Carbon Derived from Prawn Shell and Its Application for Removal of Heavy Metal Ions

**DOI:** 10.3390/ma12020241

**Published:** 2019-01-12

**Authors:** Jian Guo, Yaqin Song, Xiaoyang Ji, Lili Ji, Lu Cai, Yaning Wang, Hailong Zhang, Wendong Song

**Affiliations:** 1College of Food and Medical, Zhejiang Ocean University, Zhoushan 316022, China; guoj2019@126.com (J.G.); syq24688@163.com (Y.S.); top330327@163.com (X.J.); 2Institute of Innovation & Application, Zhejiang Ocean University, Zhoushan 316022, China; jll-gb@163.com (L.J.); lucai89@126.com (L.C.); wynzjou@126.com (Y.W.); 3College of Petrochemical and Energy Engineering, Zhejiang Ocean University, Zhoushan 316022, China

**Keywords:** prawn shell, activated carbon, specific surface area, adsorption behavior, heavy metal ion

## Abstract

The aim of this study was to optimize the adsorption performance of activated carbon (AC), derived from the shell of *Penaeus vannamei* prawns, on heavy metal ions. Inexpensive, non-toxic, and renewable prawn shells were subjected to carbonization and, subsequently, KOH-activation to produce nanoporous K-Ac. Carbonized prawn shells (CPS) and nanoporous KOH-activated carbon (K-Ac) from prawn shells were prepared and characterized by FTIR, XRD, BET, SEM, and TEM. The results showed that as-produced K-Ac samples were a porous material with microporous and mesoporous structures and had a high specific surface area of 3160 m^2^/g, average pore size of about 10 nm, and large pore volume of 2.38 m^3^/g. Furthermore, batches of K-Ac samples were employed for testing the adsorption behavior of Cd^2+^ in solution. The effects of pH value, initial concentration, and adsorption time on Cd^2+^ were systematically investigated. Kinetics and isotherm model analysis of the adsorption of Cd^2+^ on K-Ac showed that experimental data were not only consistent with the Langmuir adsorption isotherm, but also well-described by the quasi-first-order model. Finally, the adsorption behaviors of as-prepared K-Ac were also tested in a ternary mixture of heavy metal ions Cu^2+^, Cr^6+^, and Cd^2+^, and the total adsorption amount of 560 mg/g was obtained.

## 1. Introduction

Activated carbon (AC) is a remarkable adsorbent with high specific surface area, high pore volume, and adjustable surface physical and chemical properties. It is widely used in the remediation of industrial waste water and contaminated groundwater [1,2]. Heavy metal pollution is a serious environmental problem [3,4]. Copper is one of the most widely used and most common heavy metals in industrial production activities [5,6,7,8]. Excessive exposure to Cu(II) can lead to the accumulation of Cu(II) through aquaculture organisms to the human body via food chains [9,10,11]. Cr(VI) mainly exists as anions (CrO_4_^2−^, HCrO_4_^−^, and Cr_2_O_7_^2−^) in solution, and its solubility and mobility are higher than that of Cr(III) [12,13]. Furthermore, Cr(VI) is a strong carcinogenic and mutagenic agent, and its antoxicity is 100 times higher than that of Cr(III) [14]. Among heavy metals, Cd, as one of the most important persistent inorganic pollutants, can generate severe toxicity to plants, animals, and humans, even at low concentrations [15]. Thus, the removal of heavy metals from water is urgently needed. In order to enable AC to adsorb various heavy metal ions in wastewater rapidly and effectively, the internal structure and surface properties of AC can be purposefully modified and altered according to the various characteristics of the heavy metal ions to be removed and the differences between the external environment. However, the high production cost of AC is a major challenge, which has greatly hindered its practical application. In recent decades, renewable, nontoxic, and low-cost alternative bio-based materials, such as shells, agricultural residues, and other biomass [16,17,18,19] have been paid much attention. By using a very promising cost-efficient technology, researchers can turn those waste materials into carbonaceous materials, i.e., ACs. There are many studies that have been devoted to the manufacturing of high specific surface area carbon adsorbents [20,21,22,23]. Basically, two methods, i.e., physical and chemical activation, are usually employed to produce ACs [24]. It has been reported that KOH chemical activation is better than steam activation, with respect to the development of porosity, specific surface area, and total pore volume [25]. A large number of experimental studies have shown that biomass adsorbents prepared using different raw materials have good adsorption properties for various heavy metal ions. For instance, a piece of charcoal derived from coconut shell has a higher Cr adsorption capacity than that of commercial activated carbon after being oxidized with nitric acid [26]. Activated carbon with a high specific surface area of 2943 m^2^/g was reported and prepared from lignin of papermaking black liquor by KOH activation and used for the application of Ni(II) adsorption [27]. Banana peels were also subjected to carbonization and, subsequently, KOH-activation, and its maximum adsorption capacity for Cu^2+^, Ni^2+^, and Pb^2+^ was reported to follow the order: Cu^2+^ (14.3 mg/g) < Ni^2+^ (27.4 mg/g) < Pb^2+^ (34.5 mg/g) [28].

Prawn shell is a kind of food waste and industrial by-product in our daily life, which mainly contains chitin (polysaccharide), calcium protein, and crude fat, which can be converted to carbon material by simple pyrolysis [29]. In industrial prawn processing, approximately 20% of the gross weight of prawn is abandoned as waste. Most prawn shells are discarded, or made into fodder. Only a small portion can be used to extract chitin, and its preparation process leads to very serious environmental pollution [30,31,32]. Research on the use of prawn shell waste to prepare a new type of biomass carbon material with high specific surface area can not only solve the problem of ecological environment pollution, but also reduce production costs.

Biochar has attracted widespread attention due to its low cost, high specific surface area, good stability, and wide range of sources [33]. The adsorption capacity of biological carbon is mainly determined by its physical and chemical properties, such as elemental composition, surface chemistry, and structural characteristics, which are usually controlled by preparation conditions, including calcination temperature, inert gas flow, heating rate, and time [34]. Therefore, in the process of preparing carbon material with high specific surface areas, some modifiers are usually used to modify its structure and surface functional groups. Among many activated agents, KOH has been widely employed for preparing AC materials by many researchers, since it can result in ACs with defined micropore size distribution, high pore volume, and a very high specific surface area of up to 3000 m^2^/g [35,36,37]. Ru-Ling Tseng [38] produced carbonaceous adsorbents with surface areas ranging from 841 to 1221 m^2^/g by using corncobs at 780 °C for carbonization and, subsequently, KOH-activation.

In this study, the preparation of AC derived from prawn shells was designed by processing carbonization (CPS) and KOH activation (K-Ac). The adsorption performance of as-prepared adsorbent samples was investigated, and the adsorption efficiency of as-prepared K-Ac was carried out by using Cd^2+^ in aqueous solution as a model adsorbate, and the effect of pH value, initial concentration, and adsorption time on Cd^2+^ adsorption was systematically investigated. The data of these experiments were evaluated and simulated by various adsorption kinetics and thermodynamics. Finally, the adsorption properties of K-Ac in heavy metal ions Cu^2+^, Cr^6+^, and Cd^2+^ ternary systems were also studied.

## 2. Materials and Methods

### 2.1. Materials

Raw prawn shell (*Penaeus vannamei*) was provided by Zhoushan Evergreen Seafood Food Co., Ltd., Zhoushan, Zhejiang, China. It was washed repeatedly with deionized water to remove various impurities, and dried at 110 °C. A metal ion Cd^2+^ solution was prepared by using cadmium nitrate, which was purchased from China Traditional Chinese Medicine Group Chemical Reagents Co., Ltd. (Shanghai, China). Cadmium stock solution (Cd^2+^ mass concentration of 0.1 g/L) was prepared by dissolving 0.2744 g of cadmium nitrate in 1 L deionized water.

All work solutions were obtained by proper dilution with deionized water. Hydrochloric acid, sulfuric acid, phosphoric acid, potassium hydroxide, copper nitrate, potassium dichromate, and other chemicals used in this study were all of analytical reagent grade and purchased from Sinopharm Chemical Reagent Co., Ltd. (Shanghai, China). Deionized water was used throughout the experimental process.

### 2.2. Preparation of K-Ac Materials

The tubular furnace process carbonization and KOH activation method were employed for preparing K-Ac adsorbents according to the method reported previously, with some modifications [39]. Firstly, the square crucible containing 20 g of raw prawn shell was placed in a three-zone tube furnace (Guangzhou Dijiduo Instrument Co., Ltd, Guangzhou, China). Then, nitrogen was introduced at a flow rate of 100 mL/min for about 1 h, followed by a 10 °C/min heating rate. After carbonization at 800 °C for 3 h, the residue was mixed with 1 mol/L HCl for 1 h. Subsequently, the residue was filtered, washed to neutrality, and dried for 8 h at 105 °C to give the prawn shell biochar (CPS). For activation, the mass ratio of potassium hydroxide and CPS 3:1 was uniformly mixed, and ground in a mortar. The mixture was then placed in a three-temperature zone tube furnace, and activated at 800 °C for 1 h. The residue was mixed with 2 mol/L HCl for 2 h. Finally, the residue was filtered, washed to neutrality, and dried for 6 h at 105 °C to obtain the modified prawn shell biomass carbon (K-Ac). The resultant K-Ac materials were used for characterization and Cd^2+^ adsorption studies. The production steps are illustrated in Figure 1.

### 2.3. Adsorbent Characterization

Scanning Electron Microscopy (SEM, JSM-6360LV, JEOL, Japan Electronics Corporation, Beijing, China) with high resolution was employed to analyze the morphology of the adsorbent surface. A small amount of the sample with double-sided adhesive was fixed to the sample holder, and then sprayed with an ultra-thin gold film in a vacuum-coating machine. Acquired information was obtained for 100 s at 20 kV and 15 mm scan spacing. Nitrogen adsorption-desorption experiments (BET) were performed by using the Quantachrome Autosorb NOVA2200e analyzer (American Conta Instrument Co., Ltd., Shanghai, China) at 77.0 K; the specific surface area was calculated by using the BET method; the distribution of pore size was determined from the adsorption branch; and the total pore volume was determined by nitrogen adsorption at p/p0 = 0–1 to obtain nitrogen adsorption desorption isothermal curves. The elemental mapping image (EMI) was obtained by transmission electron microscopy (TEM) (JEM-2100F, Beijing Kesi Instrument Co., Ltd., Beijing, China). The DX-2700 X-ray diffractometer (XRD) (Japan Science Co., Ltd., Shanghai, China) was used to analyze the phase transition of as-prepared samples. Functional groups and the chemical bond analysis of the sample were carried out using a Fourier transform infrared spectrometer in a wave-number range of 500–4500 cm^−1^ by a resolution of 4 cm^−1^ in transmittance mode (FTIR, IR Affinity-1s, Nikon Corporation, Shanghai, China). The residual Cd^2+^ ion concentration after adsorption was determined by a UV-Vis Spectrophotometer (UV-1200, Shanghai Mapada Instruments, Shanghai, China) at 518 nm. The residual concentration of Cd^2+^, Cr^6+^, and Cu^2+^ was analyzed using an Atomic absorption spectrophotometer (TAS-990, Shanghai Mapada Instruments, Shanghai, China).

### 2.4. Adsorption Experiments

Batch mode adsorption experiments were conducted by agitating 50 mL 10 mg/L of Cd^2+^ solution and 0.25 g K-Ac in a sealed reagent bottle. All batch adsorption experiments were first oscillated for 60 min on a shaker (150 r/min) to reach the adsorption equilibrium, and then centrifuged for 5 min at 8000 r/min. The purpose of centrifugation was to separate the activated carbon from the solution, because the K-Ac is insoluble in water. After being adsorbed, it needed to be centrifugally separated from solution; the ion concentration in the solution could be measured accurately. The pH of the solutions were adjusted to the desired values, with 0.1 mol/L HCl or 0.1 mol/L NaOH solutions. Some other conditions were researched, including the effect of pH values of adsorbent solutions (pH = 1.0–7.0) using 10 mg/L of the initial Cd^2+^ concentration; the contact time (20–200 min) at pH 5.0 using 10 mg/L inital Cd^2+^ solution and took samples every 20 min; and the influence of cadmium concentration in the solution (10–400 mg/L) at pH 5.0. The Cd^2+^ equilibrium adsorption was calculated using the following equation:(1)$$q=\frac{(C_{0}-C)\cdot V}{m}$$
where q (mg/g) is the adsorption capacity values of prawn shell-based biosorbent for Cd^2+^, and C_0_ and C (mg/L) are the liquid-phase concentrations of Cd^2+^ before and after the adsorptions, respectively. V (L) is the volume of the solution and m (g) is the mass of the dry adsorbent used. The experimental results are the average values from three replicated runs.

A metal ion Cu^2+^ solution was prepared from copper nitrate and used to study the adsorption capacities of K-Ac in heavy metal ions Cu^2+^, Cr^6+^, and Cd^2+^ ternary systems. Copper stock solution (Cu^2+^ mass concentration of 0.2 g/L) was prepared by dissolving 0.5861 g of copper nitrate in 1 L deionized water; then, 10 mL of copper stock solution was transferred to a 100 mL volumetric flask to prepare a Cu^2+^ standard solution (Cu^2+^ mass concentration of 2 μg/mL). A metal ion Cr^6+^ solution, applied to determine the adsorption amounts of K-Ac in heavy metal ions Cu^2+^, Cr^6+^, and Cd^2+^ ternary systems, was potassium dichromate. Chromium stock solution (Cr^6+^ mass concentration of 0.1 g/L) was prepared by dissolving 0.2829 g of potassium dichromate in 1 L deionized water, and then, 10 mL of chromium stock solution was transferred to a 1 L volumetric flask to produce a Cr^6+^ standard solution (Cr^6+^ mass concentration of 1 μg/mL).

For the ternary system, batch mode adsorption experiments were conducted by stirring 50 mL of the desired concentration of metal ion solutions of desired concentrations in sealed reagent bottles in all of the adsorption experiments. Cu^2+^ and Cr^6+^ were added as interfering ions to solutions of various Cd^2+^ concentrations (10–400 mg/L), and then corresponding amounts of Cu^2+^ and Cr^6+^ standards were added. 0.25 g K-Ac sample was added at pH 5.0 and the solution was first oscillated for 60 min on a shaker (150 r/min) to achieve adsorption equilibrium, and then centrifuged for 5 min at 8000 r/min.

### 2.5. Thermodynamics and Kinetics of K-Ac

#### 2.5.1. Adsorption Thermodynamics

In order to study the maximum adsorption and adsorption isotherm curves of K-Ac for Cd^2+^, the two most common adsorption isotherms, Langmuir and Freundlich, were used to fit the adsorption process [40]. The Langmuir monolayer adsorption isotherm can be applied to the adsorption of heavy metals for a certain period of time, while the Freundlich adsorption isotherm model is often used to fit the interaction between adsorbed molecules on the surface of heterogeneous media. The Freundlich adsorption isotherm model is often used to fit the interaction between the adsorption and distribution on the surface of non-uniform medium, and can be used in various non-ideal conditions for the adsorption process and the multi molecular adsorption process. Therefore, the adsorption process can be well-described at an appropriate concentration. The expression formulas are given below:(2)$$Q_{e}=\frac{K_{L}Q_{m}C_{e}}{1+K_{L}C_{e}}$$
(3)$$Q_{e}=K_{F}{C_{e}}^{\frac{1}{n}}$$
(4)RL=1(1+KLCe)
where Q_e_ and Q_m_ (mg/g) are the adsorption amounts when reaching equilibrium and the monolayer adsorption capacity, respectively. C_e_ (mg/L) is the liquid phase concentration of Cd^2+^ at equilibrium. K_L_ and K_F_ (mg/g) are the Langmuir and Freundlich equilibrium kinetic models, respectively; n (dimensionless) is a constant related to the nature and the intensity of adsorption; R_L_ is a non-dimensional separation factor, which can predict whether the adsorption isotherm form is favorable to the adsorption process between the adsorbate and adsorbent (if R_L_ > 1 indicates that it is not suitable for adsorption, R_L_ = 1 means that the isotherm form is linear; 0 < R_L_ < 1 indicates that it is suitable for adsorption; and R_L_ = 0 is irrelevant [41]).

#### 2.5.2. Adsorption Kinetics

In order to study the kinetics of adsorption, the experimental curve was fitted to a quasi-first-order and quasi-second-order model. The quasi-first-order kinetic model is a Lagergren first-order rate equation based on the amount of solid adsorption. The quasi-second model is based on the assumption that the adsorption rate is controlled by the chemisorption mechanism, which involves electron sharing or electron transfer between the adsorbent and adsorbate. Both models are the difference between the amount of Cd^2+^ adsorbed by K-Ac at t-time and the amount of adsorption at equilibrium. They are the driving force for adsorption. The adsorption rate of the quasi-first-order kinetic model [42,43,44,45] is proportional to the driving force, and the adsorption rate of the quasi-second-order kinetic model [46,47,48,49] is proportional to the square of the driving force. The expression formulas are given below:(5)$$\mathrm{In}\left(Q_{e}-Q_{t}\right)={\mathrm{InQ}}_{e}-\frac{K_{1}}{2.303}t$$
(6)$$\frac{t}{Q_{t}}=\frac{1}{K_{2}Q_{e}^{2}}+\frac{t}{Q_{e}}$$
where Q_e_ and Q_t_ (mg/g) are the adsorption amounts when reaching equilibrium and at time t, respectively. K_1_ (1/min) is the first-order kinetic constant. K_2_ (mg/(g·min)) is the second-order kinetic constant.

## 3. Results and Discussion

### 3.1. Surface Pore and Morphology Analysis

Identifying the pore structure of adsorbents is the first step before designing the adsorption processes. The pore diameter of as-prepared samples can be determined from BET measurements. These characterizations are performed for PS (prawn shell) and CPS samples in order to verify the possible modification caused by the KOH treatment. The N_2_ adsorption-desorption isotherms and the pore-size distribution at 77 K are presented in Figure 2. As seen in Figure 2: (i) The adsorption isotherms demonstrate a sharp rise and convex curve at low relative pressure ratios because of strong interactions on adsorbate surfaces; (ii) immediately, there is the appearance of an isotherm inflection point and a monolayer adsorption; (iii) and then, a multilayer adsorption gradually forms and saturates the vapor pressure, with the relative pressure continuing to increase. As shown in Figure 2A, PS does not have significant adsorption capacity, and it increases slowly when the relative pressure is high (P/P_0_ ≥ 0.9). The slow growth adsorption curve of CPS indicates that there is a certain amount of micropores, and a hysteresis loop appears as the relative pressure P/P_0_ > 0.4. It suggests the presence of a large number of large pores, or it may be a pseudo-pore formed by stacking [26]. According to the classification of IUPAC (International Union of Pure and Applied Chemistry), these are type II for PS and CPS, characteristic of macroporous solids. The adsorption isotherm of K-Ac exhibits an isotherm profile of type I, as shown in Figure 2B. When P/P_0_ is very low (P/P_0_ ≤ 0.1), the amount of adsorption is directly proportional to a relative pressure, indicating that K-Ac has a larger number of micropores because of this stage, filled with a nitrogen sample. When P/P_0_ > 0.1, the curve slowly rises, but does not reach the level state due to capillary condensation of nitrogen filled with the micropore, indicating that there is a small number of narrow mesopores or macropores. Furthermore, the condensation of nitrogen can lead to a rising curve when the relative pressure reaches the saturation pressure (P/P_0_ > 0.99). Figure 2 shows that at the same diameter, the volume of pores after calcination increases, indicating that a certain number of pores are generated. Most pore sizes of K-Ac are in the range of 0–10 nm, and the peak value is close to 1.5 nm, indicating that the most developed pore structure is at 1.5 nm. In addition, the BET surface area, total pore volume, and average pore diameter of isotherm data are analyzed. The results obtained from nitrogen isotherms are presented in Table 1. As indicated in Table 1, the surface area of K-Ac is 86 times greater than that of CPS, and 835 times that of PS. All the above results indicate that developing the pore structure of K-Ac results from the KOH activation. These increases in the value of specific surface area, pore volume, and average pore radius reveal that K-Ac has stronger adsorption characteristics of heavy-metal adsorption than PS and CPS. In order to facilitate the movement of adsorbates on the porous structure of adsorbents, high values of surface area, pore volume, and average pore radius are desirable [50].

The typical SEM photographs of the as-prepared samples PS (A), CPS (B), and K-Ac (C) are shown in Figure 3. It can be seen from Figure 3A that there are some organic substances in PS which appear as a fibrous shape and may act as a binder, and other components are arranged in a layered structure. The structure of CPS is a porous microsphere-stacking structure, as illustrated in Figure 3B. In Figure 3B2, it illustrates that after a carbonization process, CPS particles appear as irregular cottony features. The distribution of pores is homogeneous, rich, and abundant, with diameters ranging from 100 to 200 nm. As can be seen from Figure 3C, the surface of K-Ac samples becomes a fluffy, porous microsphere-stacking structure, showing relatively regular pores and an increase in the number of pores, with diameters ranging from 20 to 100 nm. On the other hand, K-Ac (Figure 3C) has retained porous microspheres before activation, the crystals are significantly smaller, the nanopore structure becomes more developed, and the arrangement is more compact, which are key factors for the adsorption of the adsorbent.

After treatment with 3 M HCl, morphologies of PS, CPS, and K-Ac could be observed by TEM, as illustrated in Figure 4. The results show that these three samples appear as a spherical shape with some sporadic distribution of pores, which is in accordance with the results of SEM. As shown in Figure 4A, the number of holes on the surface of PS is relatively small, the agglomeration of holes is not very obvious, and the aperture or pore size is relatively large. As illustrated in Figure 4B, the distribution of pores is homogeneous and abundant. As indicated by the BET results, the average pore size of CPS is 3.83 nm. As exhibited in Figure 4C, K-Ac shows a relatively regular aperture and an increase in the number of pores, with a more compact arrangement to render nanoporosity, and high specific surface areas. This is probably due to the interior etching process of KOH activation [51]. It has been indicated that KOH might promote the development of new micropores induced by KOH activation [28].

### 3.2. XRD Crystalline and FTIR Phase Analysis

The crystalline structure of as-prepared samples is characterized by XRD. The XRD spectrum of CPS and K-Ac is presented in Figure 5. From Figure 5, it can be seen that almost all peaks of CaCO_3_ in PS became CaO (PDF# 78-0649), and the corresponding peaks are at 2θ = 32.203°, 37.346°, 53.854°, 64.152°, and 67.373 [52,53]. Since CaO reacts easily with H_2_O in the air and becomes Ca(OH)_2_, there is a characteristic peak of Ca(OH)_2_ (PDF# 87-0673) in the XRD pattern of the untreated adsorbent [10], and its corresponding characteristic spectra are at 2θ = 18.089°, 28.662°, 34.088°, 47.123°, 50.794°, 54.336°, 62.538°, and 64.226°. In addition, the strong diffraction peak at 26.381 may have been caused by the partial ordering of carbon, and there is a less pronounced amorphous diffraction peak around 31°, which may be residual amorphous carbon. Compared with K-Ac, the number and intensity of characteristic diffraction peaks decreases after KOH activation. The corresponding characteristic peaks are at 2θ = 23.022°, 29.405°, 35.965°, 39.401°, 43.145°, 47.489°, and 48.512°, indicating that the degree of crystallinity decreases. There is a distinct characteristic peak in the spectrum at about 2θ = 25°, and a faint characteristic peak at 2θ = 42°, which are the diffraction peaks of microcrystalline (002) and (100) crystal planes in the structure of disorder layer graphite, respectively. The diffraction peaks corresponding the (002) crystal plane are broader in the spectrum, indicating the amorphous state of K-Ac. The (100) peak of K-Ac is flat or almost disappears, and shows a diffusive shape with a large diffraction angle, which indicates that through KOH activation, K-Ac has a disordered crystallite layer, disordered carbon structure, and a smaller size of graphite-like crystallites, which may suggest strong adsorption capacity [54,55,56].

FTIR vibrational spectra of CPS and K-Ac are presented in Figure 6. For CPS, it can be seen from Figure 6 that CPS peaks at 1440 cm^–1^ and 1041 cm^–1^ are characteristic peaks of calcium carbonate, and the out-of-plane vibrational deformation peak of CO_3_^2–^ appears near 872 cm^–1^. The main intense bands of K-Ac are observed at 3640 cm^–1^, 3430 cm^–1^, 1420 cm^–1^, and 1040 cm^–1^. There are no major changes in the location of main absorption peaks, indicating their specific similar chemical structure. The peak at 3640 cm^–1^ is the characteristic infrared peak of Ca(OH)_2_. At the same time, a large number of –OH stretching vibration peaks appear at 3430 cm^–1^, representing hydroxyl groups (vibration frequencies of water, alcohols, and phenols) and N–H stretching vibrations. Two broad peaks are located in the range 1620–1380 cm^–1^, which can be assigned to the vibration of C=C bonds in aromatics and aromatic C–O stretching. The 1440 cm^–1^ peak shifts to 1420 cm^–1^, and the enhanced peak intensity at 1040 cm^–1^ is a characteristic peak of calcium carbonate. From the above analysis, it suggests that the KOH modification does not completely change existing groups on the surface of adsorbent CPS; however, after modification, the functional groups favoring metal adsorption, such as the hydroxyl groups, increase greatly, and the adsorption capacity of modified samples (K-Ac) to heavy metals is enhanced significantly. The reason for this may be that the large specific surface area, porous structure, and good adsorption properties of K-Ac are increased and promoted by KOH activation at high temperatures [57,58,59].

### 3.3. Effect of Solution pH

It is well-known that pH value is a key factor for promoting or restraining the adsorption process because it can affect the existence and the surface charge of the adsorbent [60,61]. As shown in Figure 7, the adsorption capacity of Cd^2+^ on K-Ac is significantly dependent on pH values. With the increase of pH value in the solution, the adsorption capacity of K-Ac to Cd^2+^ increases. When the pH value is 7, the adsorption amount of K-Ac is as high as 120 mg/g. The results clearly illustrate that the pH values have affected the functional groups or charges on the surface of the adsorbent. Under an acidic condition, the solution contains a large number of H^+^ ions. These H^+^ ions will occupy the adsorption sites on the surface of the adsorbent, which in turn will cause Cd^2+^ to be unable to precipitate on the surface of the adsorbent, weakening its adsorption capacity. On the other hand, the H^+^ ion concentration decreases as pH increases, the H^+^ ions do not compete with Cd^2+^ anymore, and there are more active groups and active sites available; thus, Cd^2+^ ions can obtain more OH^−^ ions in the solution. Generally, the adsorption capacity of heavy metals by an adsorbent increases with an increasing pH value. When the pH value is higher than the critical pH value of a heavy metal, hydrolysis and precipitation are dominant in the solution [62]. The critical pH value of Cd(II) ion is around 8.0 [63].

### 3.4. Effect of Contact Time and Adsorption Kinetics

The strong ability of K-Ac interacting with Cd^2+^ was desirable and beneficial with respect to real practical application. To investigate the adsorption kinetics of Cd^2+^ on K-Ac, an initial concentration of 10 mg/L was used, and results are shown in Figure 8. At the initial period (20–80 min), the adsorption capacity of Cd^2+^ on K-Ac increased rapidly, as illustrated in Figure 8A; then, it slowly increased and began to stagnate as its contact time prolonged with K-Ac. This phenomenon could be interpreted by the fact that a great number of vacant adsorption sites were available for Cd^2+^ adsorption at the initial stage, and after that, the remaining effectively adsorption sites were difficult to be occupied due to repulsive forces from occupied sites nearby or between molecules on the top surface of K-Ac, or from the bulk solution. In addition, the quasi-first-order (QF) and the quasi-second-order (QS) models were employed to further understand the characteristics of the adsorption process of Cd^2+^ on K-Ac. The QF and the QS models were fitted with the adsorption kinetic data of Cd^2+^ on K-Ac, and the curves are shown in Figure 8B and Figure 8C, respectively. The kinetic parameters and correlation coefficients (R2) determined from these two models are listed in Table 2. We noticed that the experimental data closely followed the nonlinear regression of QF, and the R2 values (0.99587) of the QF kinetic model was much higher than that of QS (0.91054), suggesting that the adsorption process of Cd^2+^ on K-Ac was more suitable for the QF kinetic model.

### 3.5. Effect of Initial Cd^2+^ Concentration and Adsorption Isotherms

The effect of initial Cd^2+^ concentration on the adsorption capacity of K-Ac was investigated by adsorption experiments, and the results obtained are presented in Figure 9. From Figure 9A, the adsorption capacity of Cd^2+^ adsorbed on K-Ac was enhanced with the increase of initial Cd^2+^ concentration from 10 to 400 mg/L and reached up to 258 mg/g, and there was a trend of further increase. These results showed that, at lower initial Cd^2+^ concentrations (10–80 mg/L), intense competition for Cd^2+^ under aqueous conditions caused these ions to be condensed to a K-Ac host system with a reticulated porous structure and a rich OH^−^ group on the surface, and the adsorption rate increased rapidly. When the initial concentration of Cd^2+^ increased (80–400 mg/L), the adsorption rate gradually decreased, where the decrease in adsorption rate may have been due to the limited number of active sites on the K-Ac surface.

In order to further interpret the experimental adsorption data, the adsorption isotherm characteristics were also employed to investigate the adsorption behaviors of the heavy metals onto the prawn shell-derived activated carbon. Among a number of isotherm models, Langmuir and Freundlich’s isotherms are the most commonly used ones for describing the adsorption of heavy metals and organic compounds on ACs [64]. The adsorption fitting curves of the two models are described in Figure 9B,C. According to the R^2^ values of the two isotherm models listed in Table 3, it was found that the experimental results could be best fitted by the Langmuir model, as shown as Figure 9B. It was illustrated that this adsorption process was homogeneous, and the active sites on the K-Ac surface were evenly distributed. The value of the dimensionless separation factor R_L_, 0.862, was in the range of 0 < R_L_ <1, indicating that it was within the range of the concentration studied. In addition, the n value of the Freundlich isotherm model constant was 1.011, within the range of 1–7, suggesting that the K-Ac is beneficial to the adsorption of Cd^2+^ after activation.

### 3.6. Adsorption Capacity in Cu^2+^, Cr^6+^, and Cd^2+^ Ternary System

The adsorption capacities of K-Ac in the Cu^2+^, Cr^6+^, and Cd^2+^ ternary system and in their single systems are shown in Figure 10. The experimental results illustrated that the adsorption performance of K-Ac for metal ions in the multi-component heavy metal ion system was different from those in a single system. The adsorption amount of metal ions in single systems and the total adsorption amount of metal ions in a ternary system are shown in Figure 10A. As the initial concentration of the three metal ions increases, the adsorption amount of K-Ac gradually reaches equilibrium, and the order of sorption is Cr(VI) (318 mg/g) > Cu (280 mg/g) > Cd(II) (256 mg/g). As shown in Figure 10A, for the ternary system after equilibrium, the total adsorption amount of K-Ac for the three ions is 560 mg/g, which is about two times greater than that in the single system, but not the superimposed sum of ions adsorbed in their single ion system. It is suggested that for the adsorption experiments in the ternary system, there would be a competitive relationship among the three ions, which will affect the adsorption of metal ions by K-Ac. Furthermore, in the ternary system, as the initial concentration of the metal solution increases, the adsorption amount of Cd^2+^ by K-Ac was almost unchanged, while the adsorption capacity of Cr^6+^ was greatly decreased, as shown in Figure 10B, which is the individual adsorption amount of each metal in a single system and a ternary system. Compared with the single system, the adsorption amount of Cd^2+^ by K-Ac in the ternary system was slightly reduced in the same condition, indicating that the presence of Cu^2+^ and Cr^6+^ had little effect on the adsorption of Cd^2+^ by K-Ac. The adsorption capacity of K-Ac for Cr^6+^ in the ternary system was lower than that in the single system, suggesting that the presence of Cu^2+^ and Cd^2+^ had a strong antagonistic inhibition effect on the adsorption of K-Ac to Cr^6+^. Meanwhile, the difference in the adsorption capability of K-Ac to each heavy metal ion can be expressed by the ratio of adsorption amounts, i.e., Q3/Q1 (where Q3 is the adsorption capacity of K-Ac to a certain heavy metal ion in the ternary system; Q1 is the adsorption capacity of K-Ac to corresponding metal ions in a single system). The Q3/Q1 ratio of Cd^2+^ was approximately equal to 1, and the value of Cr^6+^ was much less than 1, which indicated that the adsorption of Cd^2+^ by K-Ac was not affected by interfering ions Cu^2+^ and Cr^6+^, while by the presence of Cd^2+^ and Cu^2+^, the adsorption of Cr^6+^ by K-Ac was inhibited to some extent; thus, the adsorption amount of Cr^6+^ was reduced. However, the total adsorption capacity of K-Ac for three metal ions in the ternary system could reach up to 560 mg/g, which was higher than that of the single heavy metal ion. From these data, it can be inferred that in the ternary system, the adsorption capacity of copper becomes the lowest due to the presence of Cd^2+^ and Cr^6+^ ions, and the adsorption capacity of copper drops from 280 to 24 mg/g.

The possible sorption mechanism on bioadsorbents can be explained by interactions between the heavy metal pollutant ions and the functional groups present on the adsorbent surface. Among various factors that affect the biosorption preferences of a sorbent, binding of metal ions on biomaterials largely depends on the physicochemical properties of metals [65]. It has been reported that in general, the greater the atomic weight, electronegativity, electrode potential, and ionic size, the greater the affinity for sorption will be [66]. The order of sorption will be Cd(II) > Cu(II) > Cr(VI), according to atomic weight. In the single system, the adsorption amount of K-Ac to Cr^6+^ is higher than that to Cu^2+^ and Cd^2+^, as shown in Figure 10A. In accordance with electronegativity, the larger the radius of metal ions, the smaller the electronegativity. Therefore, the order of sorption will become Cr(VI) > Cu(II) > Cd(II) [67]. The obtained results are consistent with the physicochemical properties of metal ions under this study. Compared with the adsorption amount in the single system under the same condition, the adsorption capacity of K-Ac to Cd^2+^ and Cr^6+^ in the ternary system is reduced, while the adsorption capacity of K-Ac to Cr^6+^ decreases greatly, which can be ascribed to the overlapping of adsorption sites of respective metal ions. There exist a variety of binding sites on the K-Ac biomass that are partially specific for individual metal species [68]. The removal efficiency of the same adsorbent in a multi-component solution is normally lower than that of single-component ones, due to the less availability of binding sites [69]. Metal ions have been revealed to coordinate in adsorbent predominantly to carboxylic moieties without excluding hydroxyl groups [67]. Thus, in the present trimetallic combination, the sorption of metal ions is a competitive process between ions in solution and those sorbed onto the biomass surface. The results of this study indicate that the presence of Cu^2+^ and Cr^6+^ has little effect on the adsorption of Cd^2+^ by K-Ac, while Cu^2+^ and Cd^2+^ have strong antagonistic inhibition on the adsorption of K-Ac to Cr^6+^.

## 4. Conclusions

ACs with a higher specific area and excellent Cd^2+^ adsorption performance have been successfully prepared from prawn shell waste through high-temperature carbonization and KOH activation. The resultant activated carbons had alarge specific area of 3159.65 m^2^/g and nanoporosity with narrow pore size distribution and lower crystallinity, which presented more advantages for adsorption purposes than PS and CPS, and its specific surface area was also 835 and 86 times higher than PS and CPS, respectively. After treatment and enhancement, the adsorption capacity of K-Ac to Cd^2+^ could reach up to 256 mg/g, and there was a trend of further increase. An analysis of the kinetic and isotherm models for the adsorption of Cd^2+^ on K-Ac indicated that the experimental data were well-described by the quasi-first-order model and Langmuir isotherm. Furthermore, the adsorption properties of K-Ac in a ternary mixture of heavy metal ions Cu^2+^, Cr^6+^, and Cd^2+^ were also examined. The sorption preferences of K-Ac were found, that binding of metal ions on biomass largely depended on the physicochemical properties of the metals. For the ternary system after equilibrium, the maximum adsorption capacity of K-Ac at 25 °C was 560 mg/g. In summary, it is expected that the inexpensive K-Ac could be used as a promising bioadsorbent to remove heavy metal ions from industrial wastewater.

## Figures and Tables

**Figure 1 materials-12-00241-f001:**
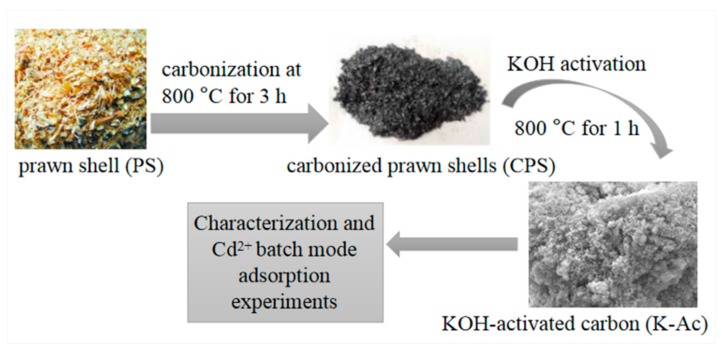
Diagram of the K-Ac preparation process.

**Figure 2 materials-12-00241-f002:**
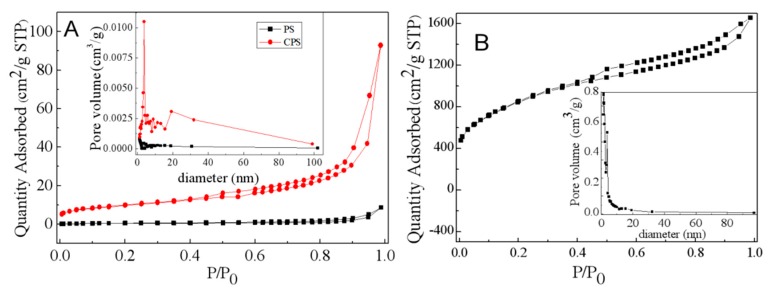
Nitrogen adsorption-desorption isotherm and pore size distribution of (**A**) PS and CPS; (**B**) K-Ac.

**Figure 3 materials-12-00241-f003:**
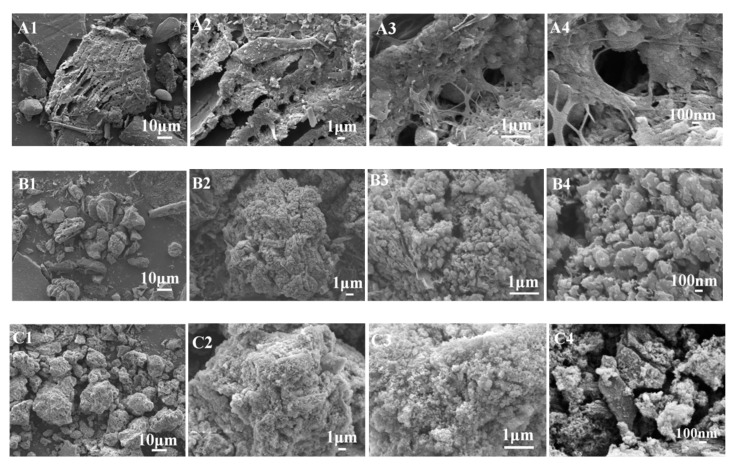
Observations with SEM: (**A**) PS (A1–A4: magnified ×5000, ×10,000, ×20,000, ×50,000, respectively), (**B**) CPS (B1–B4: porous microsphere-stacking structures of CPS with different magnification), and (**C**) K-Ac (C1–C4: nanopore structures of K-Ac with different magnification).

**Figure 4 materials-12-00241-f004:**
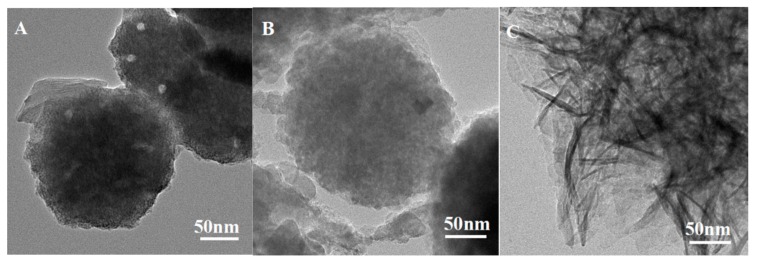
TEM images of (**A**) PS, (**B**) CPS, and (**C**) K-Ac.

**Figure 5 materials-12-00241-f005:**
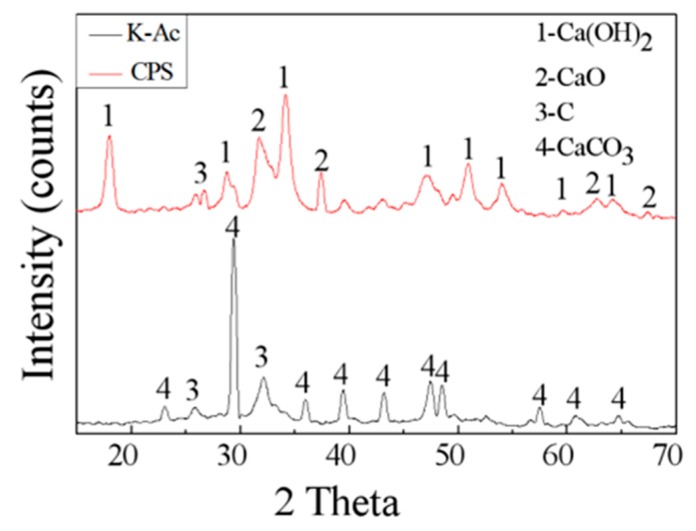
XRD spectra of as-prepared samples, CPS and K-Ac.

**Figure 6 materials-12-00241-f006:**
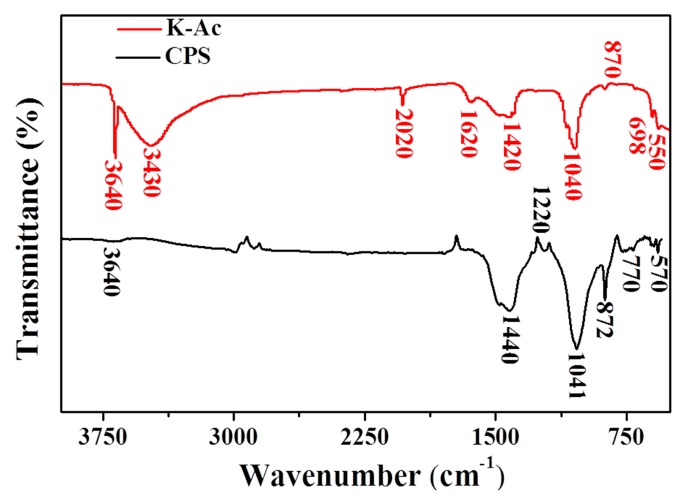
FTIR spectra of CPS and K-Ac.

**Figure 7 materials-12-00241-f007:**
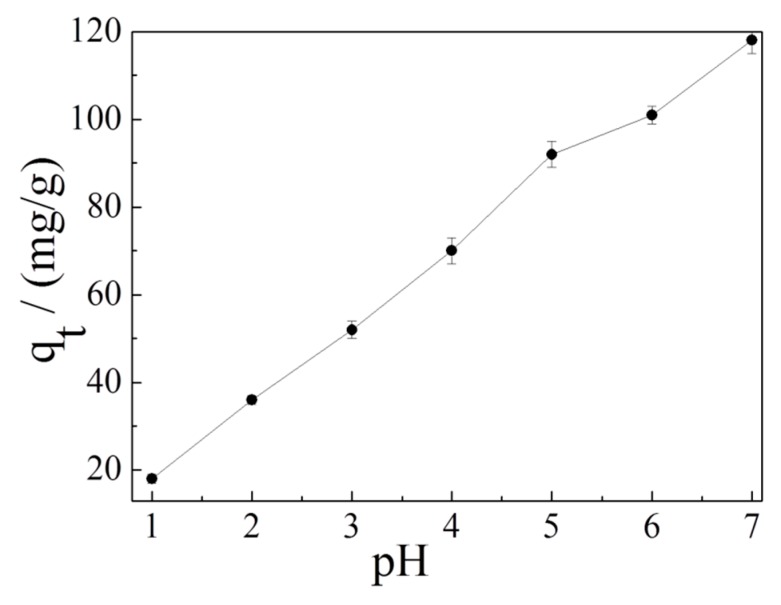
Effect of pH on the adsorption of Cd^2+^ ion on K-Ac.

**Figure 8 materials-12-00241-f008:**
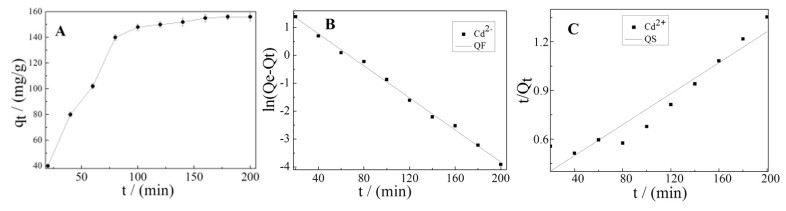
(**A**) Effect of contact time for Cd^2+^ adsorption by K-Ac; (**B**) adsorption kinetics of Cd^2+^-fitted quasi-first-order rate model; (**C**) adsorption kinetics of Cd^2^-fitted quasi-second-order rate model.

**Figure 9 materials-12-00241-f009:**
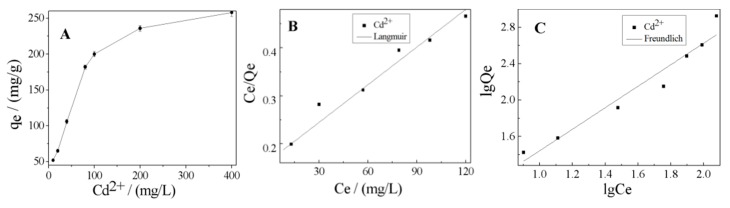
(**A**) Effect of initial concentration for Cd^2+^ adsorption by K-Ac; (**B**) Langmuir; (**C**) Freundlich.

**Figure 10 materials-12-00241-f010:**
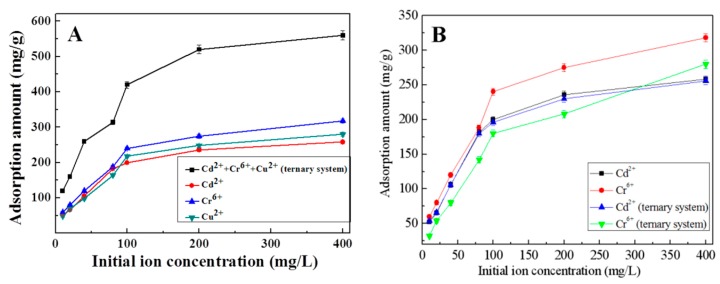
Adsorption capacity of K-Ac examined in heavy metal ion Cu^2+^, Cr^6+^, and Cd^2+^ solution; (**A**) total adsorption amount of elements in ternary systems and single adsorption capacity of each ion in single system; (**B**) single-element adsorption amount of Cd^2+^ and Cr^6+^ in different systems.

**Table 1 materials-12-00241-t001:** The specific surface area parameter of as-prepared samples.

Sample	Surface Area (m^2^/g)	Pore Volume (cm^3^/g)	Average Pore Radius (nm)
PS	4	0.014	1.349
CPS	37	0.143	3.831
K-Ac	3160	2.382	1.351

**Table 2 materials-12-00241-t002:** The kinetic parameters of the quasi-first-order (QF) and quasi-second-order (QS) models on the adsorption of Cd^2+^.

Sample	QF		QS	
K-Ac	K_1_/(1/min)	R^2^	K_2_/(mg/g·min)	R^2^
	0.066	0.99587	19.723	0.91054

**Table 3 materials-12-00241-t003:** Adsorption isotherm model parameters.

Sample	Langmuir			Freundlich		
K-Ac	K_L_	R^2^	R_L_	n	K_F_	R^2^
	0.016	0.9737	0.862	1.011	1.829	0.93716

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
