# Peer review of "Preparation and Characterization of Nanoporous Activated Carbon Derived from Prawn Shell and Its Application for Removal of Heavy Metal Ions"

_materials, 2019, doi:10.3390/ma12020241_

Reviewer 1 Report

This manuscript reports the preparation of activated carbon from prawn shells and using KOH activation. Materials were characterized by N2 adsorption, SEM, TEM, XRD and FTIR, and activated carbon was then tested for adsorption of Cd II from aqueous solution. Sorbent adsorption capacity was tested at various pHs, and compared to other metal species that might compete with Cd II. Results are interesting, and manuscript has previously been revised as parts of it appear in red color. I recommend its publication after additional minor revisions on the N2 characterization. My comments are:

p.p1 {margin: 0.0px 0.0px 0.0px 0.0px; font: 14.0px Helvetica; color: #000000} span.s1 {font: 11.7px Helvetica}

1. Minor language improvements are recommended throughout text. 

2. Use of decimal places for calculated BET surface areas is not recommended, as variability in this model is in the order of +/-10%.

3. Isotherms are in Figure 1, not in Fig. 3. Figures 2 and 3 are also called by the wrong number in the text. Finally, text should reference each figure exactly as caption appears, i.e. "Figure 1" rather than Fig. 1"

4. Furthermore, these are type II for PS and CPS, characteristic of macroporous solids. The isotherm for K-Ac is of type I, with high gas uptake at low relative pressures, thus indicating large amounts of micropores. The latter isotherm exhibits some small gas uptake at higher relative pressures due to the presence of some mesopores. The hysteresis loop for the K-Ac isotherm is of typoe H-4, indicating slit-like mesopores formed between carbon particle aggregates. The calculated pore size distribution (PSD) confirm that some mesopores indeed exist. It is recommended that authors limit the PSDs to 50nm, as N2 adsorption is only able to probe micropores and mesopores. What equation is used for calculating PSDs? What is the range used for calcualting BET specific surface areas, and what p/p0 is used for calculating pore volumes? Recommended reference for N2 adsorption is Chem. Mater.200113 (10), pp 3169–3183.

Author Response

A1: Yes, improvements have been made accordingly.

A2: Yes, thanks for your suggestions, and the numbers of calculated BET surface areas have been revised accordingly in the table and in the text.

A3: Yes, thank you for your suggestion. We have corrected the error in the text, please check the manuscript.

A4: The pore size distribution and specific surface area are measured by the isothermal adsorption characteristic curve of nitrogen: when P/P0 is in the range of 0.05~0.35, the adsorption amount and (P/P0) are in accordance with the BET equation, which is the basis for determining the specific surface area of the powder material. When P/P0>0.4, nitrogen starts to agglomerate in the micropores, pore volume and pore size distribution can be measured due to capillary aggregation.

In the first step, the nitrogen pressure starts from P0 and falls to P1. At this time, the pore condensate of the pore size from rC0 to rC1 is desorbed, and the average pore diameter of the pore region is C1, then the pore center of the pore region, the volume (VC1), the actual pore volume (VP1) and the surface area of the pore (SP1) can be obtained from the following equations:

Rk = -0.414 log(P/P0)..........................................(1)

rK is called the Kelvin radius, it depends entirely on the relative pressure P/P0, that is, the radius of the hole that starts to agglomerate at a certain P/P0, and the condensed liquid in the hole of radius rK will be vaporized and desorbed. 

Before the agglomeration phenomenon, there is already a layer of nitrogen adsorption film on the capillary arm, and its thickness (t) is also related to the relative pressure (P/P0). The Herschel equation gives this relationship:

t = 0.354[-5/ln(P/P0)]1/3 .......................................(2)

VC1 = p C1 2 L1......................................................(3)

Vp1 = p p1 2 L1 ...................................................(4)

Sp1 = 2 p C1 L1 ................................................(5)

In the above three formulas, L1 is the total length of the holes, and rC1 and rp1 can be obtained by Kelvin equation (1) and Hercept equation (2). After dividing (4) by (3) and (5) divided by (4), the volume and surface area of the pores in the first pore zone are:

VP1 =( P1/ C1)2. VC1..........................................(6)

SP1= 2 VP1/P1 ...................................................(7)

Equations (6) and (7) show that when the pressure drops from P0 to P1, the amount of nitrogen desorbed from the sample, and then this gas nitrogen is converted into the volume Vc1 of liquid nitrogen, and the size r0 to r1 can be obtained.

In the second step, the partial pressure of nitrogen is reduced from P1 to P2. At this time, the desorbed nitrogen includes two parts: the first part is nitrogen desorbed from the pore center of the rp1 to rp2 pore region, and the second part is the previous one. The nitrogen in the adsorption layer remaining in the pores of the pore region (rp0 to rp1) is desorbed by the decrease in the layer thickness (Δt2 = t1 - t2), so the volume of nitrogen desorbed in the second pore region is VC2. , pore volume (VP2) and pore area (Sg2) are:

VC2 = p C2 2 L2 + Sp1t2 .......................................(8)

Vp2 = p p2 2 L2 .............................................(9)

Sp2 = 2 p C2 L1 .............................................(10)

After the simple treatment, the pore volume and specific surface area of the second pore zone are:

VP2=( p2/C2)2[VC2- SP1t2] .................................(11)

SP2=2VP2/p2 ..........................................(12)

VC2 in the above formula (11) is a volume in which the nitrogen desorbed from the solid surface after the pressure changes from P1 to P2 and is converted into a liquid.

Similarly, when the i-th hole region is rp(i-1)~rpi, the hole volume ΔVpi and the surface area ΔSpi of the hole region are:

ΔVpi = ( pi / ci) 2 [ Vci - 2 Δ ti Δ Vpj / pj ] ... ... (13)

Spi =2Vpi/ pi ..........................................(14)

The physical meaning of equation (13) is very clear. ΔVpi is the i-th hole area, ie the volume of the hole with a hole radius from rp(i-1) to rpi, and Vci is the relative pressure from P(i-1). The amount of nitrogen desorbed from the solid surface when reduced to Pi is converted into the volume of liquid nitrogen. The formula for converting liquid nitrogen into liquid nitrogen is as follows:

V liquid = 1.547×10-3V gas..........................................(15)

The last item is the desorption nitrogen caused by Δti in the pore larger than rpi, which does not belong to the nitrogen removed from the i-hole region and needs to be deducted from Vci; (pi/ci) 2 is a coefficient which gives the radius. The pore volume for c is converted to the pore volume of p. When the pore size is small, the amount of gas desorption caused by Δt cannot be approximated as a plane, and this should be appropriately corrected. This is the commonly used DJH method.

Based on the above theoretical analysis, firstly, the P/P0 values of several experiments are designed (that is, divided into several hole areas). According to P/P0, all the parameters required for calculation can be obtained by the following related formula:

Rci = rki = -0.414 / log(P/P0)i (Kelvin equation)

Ti = 0.354[-5 / ln(P/P0)i]1/3 (Hersey equation)

Rpi = rci + ti

Pi = 1/2 (rp(i-1)+ rpi)

Ci = 1/2 (rc(i-1)+ rci)

ti = t(i-1)-ti

Ri = (pi /ci)2

Reviewer 2 Report

Overall the manuscript covers a series of interesting batch experiments with activated carbon derived from Prawn Shell. There are some unclear part on M&M (kinetics and ternary adsorption part) and results and discussion (pH effect and ternary adsorption). Clarification and revision on those parts will make readers to read the article more effectively.

L33 remarkably adsorbent -> remarkable adsorbent

L34 in the repair and 34 remediation of industrial -> in the remediation of industrial

L51 those seemingly waste materials -> those waste materials

L62-63 Please consider revising the sentence. For instance, “And banana peels was treated at 500 °C for chemical activation of KOH and its maximum adsorption capacity for Cu2+, Ni2+ and Pb2+ were in Cu2+ (14.3 mg/g) < Ni2+ (27.4 mg/g) < Pb2+ (34.5 mg/g)”. Polish further as needed.

L103 through- out -> throughout

L110 Carbonization at 800 °C for 3 h -> After Carbonization at 800 °C for 3 h?? Please clarify.

L105-117 Preparation of K-Ac materials -> Suggest to add a diagram and/or experimental steps (include pics if available) to produce the K-Ac materials, so that other researcher can clearly see the production steps.

L119-122 Suggest to revise to make complete sentences.

L137 agitating 50 mL, Cd2+ solution of desired concentrations -> agitating 50 mL of Cd2+ solution at desired concentrations. Suggest to specify the concentrations here. Also, solid-to-solution ratio should be mentioned in this section.

L152 3 duplicate runs -> 3 replicated runs

L167 Please explain why pH 5 (initial solution?) was chosen for the equilibration.

L180 Therefore, the adsorption process can be well described at an appropriate concentration. -> Is this for Freundlich model? Not clear what appropriate concentration means here.

L181 Check Qm -> Does this correspond to maximum adsorption capacity of a metal?

L191 quasi –first order and quasi-second-order model -> Suggest to explain the models briefly.

L192-193 Both models are the difference between the amount of Cd2+ adsorbed by K-Ac at t-time and the amount of adsorption at equilibrium.  -> Both models explain? Please clarify.

L193 They are the driving force for adsorption. -> Not sure what this means. Please clarify.

L205 Figure 3 -> Figure 1??

L225-227 Suggest to delete the sentence as Table 1 shows all the results.

L230 stronger characteristics?? Please clarify (specific surface area??).

L237 Figure numbers should be checked throughout manuscript. Fig 4 -> Fig 2??

L242-243 abundant with diameter ranges from 100 to 200 nm, which is conducive and beneficial to the penetration of activators -> Is this finding from current study? Otherwise, please cite reference.

L309-313 These sentences are for M&M. Suggest to delete.

L317-323 Suggest to author to check solubility of the metals tested and chemistry references. At the pH range tested (1-7), precipitation may not occur. The increased metal adsorption with increasing pH may be explained by surface charge change rather than precipitation.  

L326 adsorption kinetics -> In M&M, suggest to explain experimental procedures for the kinetics. In Fig. 7, there are 10 different data points which indicate 10 different subsampling of the samples. Or did authors included 10 times more samples to begin with and sacrifice the samples between time intervals.

L369 Describe which adsorption isotherm model was fitted better.

L405 Figure 9 -> Need clarification how (a) and (b) were divided. (b) shows individual heavy metal ion in the ternary system? Please clarify.

L439 the experimental data were well described by the quasi-first-order model and Langmuir isotherm. -> From Table 3, R2 was higher with the Freundlich model. Please clarify.

Author Response

L33 remarkably adsorbent -> remarkable adsorbent 
A: Thanks, the change has been made.

L34 in the repair and 34 remediation of industrial -> in the remediation of industrial 
A: Yes, the change has been made.

L51 those seemingly waste materials -> those waste materials 
A: Yes, the extra word has been deleted.

L62-63 Please consider revising the sentence. For instance, “And banana peels was treated at 500 °C for chemical activation of KOH and its maximum adsorption capacity for Cu2+, Ni2+ and Pb2+ were in Cu2+ (14.3 mg/g) < Ni2+ (27.4 mg/g) < Pb2+ (34.5 mg/g)”. Polish further as needed.

A: Yes, thank you for your suggestion; we have revised this sentence, please refer to the manuscript.

L103 through- out -> throughout 
A: Yes, the error was corrected.

L110 Carbonization at 800 °C for 3 h -> After Carbonization at 800 °C for 3 h?? Please clarify.

A: Yes, thanks, the corresponding correction has been made in the manuscript. This sentence means pickling after carbonization, so it should be “After Carbonization at 800 °C for 3 h”.

L105-117 Preparation of K-Ac materials -> Suggest to add a diagram and/or experimental steps (include pics if available) to produce the K-Ac materials, so that other researcher can clearly see the production steps.

A: Yes, thank you for your suggestion. We have added the diagram of K-Ac preparation process in Section 2.2. Please refer to the manuscript.

L119-122 Suggest to revise to make complete sentences. 

A: Yes, thanks for your suggestion, we have modified the sentences in this section in the manuscript.

L137 agitating 50 mL, Cd2+ solution of desired concentrations -> agitating 50 mL of Cd2+ solution at desired concentrations. Suggest to specify the concentrations here. Also, solid-to-solution ratio should be mentioned in this section.

A: Yes, thank you for your suggestion, we have added specific Cd2+ concentration and solid-liquid ratio in this part of the manuscript.

L152 3 duplicate runs -> 3 replicated runs  
A: Yes, the error was corrected.

L167 Please explain why pH 5 (initial solution?) was chosen for the equilibration.

A: Here are the reasons for choosing pH5 as the initial solution: First, according to our preliminary experiment, after adding heavy metal ion solution in water, the pH value of the solution is generally around 5; Second, our all batch adsorption experiments were used pH5 as the initial solution; Finally, the pH of wastewater contaminated by heavy metals is generally around 5, which is to better simulate the environment of wastewater.

L180 Therefore, the adsorption process can be well described at an appropriate concentration. -> Is this for Freundlich model? Not clear what appropriate concentration means here.

A: Yes, this sentence describes the Freundlich adsorption isotherm model because it can be used in adsorption processes and multimolecular adsorption processes under various non-ideal conditions. As long as the concentration of the adsorbed material in the solution is appropriate and can be observed to be adsorbed, the Freundlich adsorption isotherm model can be used to fit the adsorption process.

L181 Check Qm -> Does this correspond to maximum adsorption capacity of a metal?

A: As described in 2.5.1, Qm here does not refer to the maximum adsorption capacity of the metal, but the monolayer adsorption capacity when reaching equilibrium

L191 quasi –first order and quasi-second-order model -> Suggest to explain the models briefly.

A: Yes, thank you for your suggestion, we have added an explanation of the quasi-first-order kinetic model and the quasi-second-order kinetic model accordingly.

L192-193 Both models are the difference between the amount of Cd2+ adsorbed by K-Ac at t-time and the amount of adsorption at equilibrium.  -> Both models explain? Please clarify.

A: The quasi-first-order kinetic model is a first-order kinetic model based on solid-phase adsorption performance, while the quasi-second-order kinetic model can predict the entire adsorption process.

L193 They are the driving force for adsorption. -> Not sure what this means. Please clarify.

A: Thanks, the so-called driving force refer to the adsorption amount (Qtof heavy metal ions by K-Ac at time t. The driving force generally refers to the affinity/interaction between adsorbate and adsorbent, i.e., the interaction force affecting the adsorption process.

L205 Figure 3 -> Figure 1?? 
A: Yes, it’s a typing error, and has been fixed.

L225-227 Suggest to delete the sentence as Table 1 shows all the results.

A: Yes, thank you for your suggestion. We have deleted the corresponding sentence, please refer to the manuscript.

L230 stronger characteristics?? Please clarify (specific surface area??).

A: Thank you for your questions, because K-Ac has a higher specific surface area, a larger pore volume and a smaller pore size, the "stronger feature" here means that it has better adsorption characteristics for heavy metals than CPS and PS. We have added “adsorption” in the text.

L237 Figure numbers should be checked throughout manuscript. Fig 4 -> Fig 2??

A: Yes, it’s a typing error, and has been fixed. Please refer to the manuscript.

L242-243 abundant with diameter ranges from 100 to 200 nm, which is conducive and beneficial to the penetration of activators -> Is this finding from current study? Otherwise, please cite reference.

A: Thank you for your suggestion. Within a certain range, the larger the pore size, the better the penetration of the activator. However, there is no research exactly report, we believe that the pore diameter is in the range of 100-200 nm, and it may be beneficial to the penetration of activator. Here, we choose to delete this sentence instead.

L309-313 These sentences are for M&M. Suggest to delete.

A: Thanks, here we have deleted these sentences.

L317-323 Suggest to author to check solubility of the metals tested and chemistry references. At the pH range tested (1-7), precipitation may not occur. The increased metal adsorption with increasing pH may be explained by surface charge change rather than precipitation.  

A: Yes, in the manuscript we have added the corresponding literature and revised the sentence accordingly, please refer to the manuscript.

L326 adsorption kinetics -> In M&M, suggest to explain experimental procedures for the kinetics. In Fig. 7, there are 10 different data points which indicate 10 different subsampling of the samples. Or did authors included 10 times more samples to begin with and sacrifice the samples between time intervals.

A: Yes, thanks for your suggestion. The adsorption kinetics model experiments were carried out according to the single factor experiment of time. We have already added the corresponding explanation in the time batch experiment of Section 2.4, please refer to the manuscript.

L369 Describe which adsorption isotherm model was fitted better.

A: In this line, we have described that the Langmuir adsorption isotherm model can be better used to fit the adsorption process. The sentences are as follows:”According to the R2 values of the two isotherm models listed in Table 3, it was found that the experimental results could be best fitted by the Langmuir model” (Due to the R2 value of Langmuir is greater than that of Freundlich.).

L405 Figure 9 -> Need clarification how (a) and (b) were divided. (b) shows individual heavy metal ion in the ternary system? Please clarify.

A: (Due to one figure was added, Figure 9 now became Figure 10) In Figure 10A, it illustrated that the total amount of adsorption in the ternary system was difference from the sum of the adsorption amounts of Cu2+, Cd2+ and Cr6+ in a single ion system; In Figure 10A, these results suggested the presence/existence of competitive adsorption in the ternary system. However, the results in Figure 10B indicated that the abilities of K-Ac to adsorb Cd2+ and Cr6+ in different systems were different.

Figure 10B illustrated that individual heavy metal ion in the ternary system was different from its single system. The adsorption capacity of Cd2+ and Cr6+ in different systems was used to compare the ability of K-Ac to adsorb Cd2+ and Cr6+, and the adsorption of cadmium and chromium by the presence of interfering ions was also discussed.        

L439 the experimental data were well described by the quasi-first-order model and Langmuir isotherm. -> From Table 3, R2 was higher with the Freundlich model. Please clarify.

A: In Table 3, the R2 of Langmuir isotherm is 0.9737, and the R2 of Freundlich isotherm is 0.93716, so the adsorption process can be better described by the Langmuir isotherm.